# Topical tacrolimus for the treatment of secondary lymphedema

Jason C. Gardenier[1,*], Raghu P. Kataru[1,*], Geoffrey E. Hespe[1], Ira L. Savetsky[1], Jeremy S. Torrisi[1], Gabriela D. García Nores[1], Dawit K. Jowhar[2], Matthew D. Nitti[1], Ryan C. Schofield[3], Dean C. Carlow[3] & Babak J. Mehrara[1]

Secondary lymphedema, a life-long complication of cancer treatment, currently has no cure. Lymphedema patients have decreased quality of life and recurrent infections with treatments limited to palliative measures. Accumulating evidence indicates that T cells play a key role in the pathology of lymphedema by promoting tissue fibrosis and inhibiting lymphangiogenesis. Here using mouse models, we show that topical therapy with tacrolimus, an anti-T-cell immunosuppressive drug, is highly effective in preventing lymphedema development and treating established lymphedema. This intervention markedly decreases swelling, T-cell infiltration and tissue fibrosis while significantly increasing formation of lymphatic collaterals with minimal systemic absorption. Animals treated with tacrolimus have markedly improved lymphatic function with increased collecting vessel contraction frequency and decreased dermal backflow. These results have profound implications for lymphedema treatment as topical tacrolimus is FDA-approved for other chronic skin conditions and has an established record of safety and tolerability.

[1] Division of Plastic and Reconstructive Surgery, Department of Surgery, Memorial Sloan Kettering Cancer Center New York, New York, New York 10065, USA. [2] Weill Cornell Medical College New York, New York, New York 10065, USA. [3] Department of Laboratory Medicine, Memorial Sloan Kettering Cancer Center, New York, New York 10065, USA. * These authors contributed equally to this work. Correspondence and requests for materials should be addressed to B.J.M. (email: mehrarab@mskcc.org).

Secondary lymphedema (henceforth referred to as lymphedema) is a chronic debilitating disease that in the United States and Western countries occurs most frequently as a complication of cancer treatment. It is estimated that as many as one in three patients who undergo lymph node dissection will go on to develop lymphedema, and conservative estimates suggest that as many as 50,000 new patients are diagnosed annually[1,2]. Because lymphedema is a life-long disease with no cure, the number of affected individuals is increasing annually with current estimates ranging between 5 and 6 million Americans[3]. It is likely that this number will continue to increase in the future since the development of lymphedema is nearly linearly related with cancer survivorship, and because the prevalence of known risk factors for lymphedema, such as obesity and radiation, is rising[4].

Lymphedema is disfiguring and debilitating; patients have chronic swelling of the affected extremity, recurrent infections, limited mobility and decreased quality of life[5]. In addition, once lymphedema develops it is usually progressive. Existing approaches for lymphedema management are palliative in nature, relying on compression garments and physical therapy to decrease fluid accumulation and progression of disease. Currently, there is no known pharmacologic therapy that can halt progression or promote resolution of lymphedema[6].

Fibrosis is a clinical hallmark of lymphedema and recent studies have suggested that lymphedema may represent end stage fibrotic organ failure of the lymphatic system[7]. For example, histological studies from clinical biopsy specimens and animal studies have shown that collecting lymphatic vessels in lymphedematous limbs become progressively sclerosed and obliterated because of proliferation of smooth muscle cells and the loss of lymphatic endothelial cell lining[8,9]. Other studies have shown that the severity of lymphedema correlates with increasing subcutaneous and dermal fibrosis[10]. Finally, experimental studies using surgical models of lymphatic injury and radiation therapy have suggested that fibrosis is an independent regulator of lymphatic function and regeneration[11–13].

More recently, we have shown that CD4$^+$ cells play a crucial role in the regulation of fibrosis in both clinical and animal models of lymphedema. For example, we have found that clinical biopsy specimens of lymphedema and animal models of lymphedema are infiltrated by CD4$^+$ cells, and that the number of these cells correlates with the degree of fibrosis and clinical severity of lymphedema[14]. Patients with late-stage lymphedema had significantly more infiltrating T cells in general, specifically more CD4$^+$ cells, than those with early-stage disease. Improvements in clinical symptoms of lymphedema after lymphovenous bypass, a procedure in which obstructed lymphatics are shunted to the venous circulation, is associated with decreased tissue fibrosis and decreased CD4$^+$ cell infiltration[15]. The CD4$^+$ cell response in lymphedema, similar to other fibroproliferative disorders, is characterized by a mixed Th1/Th2 cell population[14]. Depletion of CD4$^+$ cells (but not other inflammatory cell types including CD8$^+$ cells or macrophages) or inhibition of Th2 differentiation markedly decreases the degree of fibrosis, increases lymphangiogenesis and lymphatic fluid transport, and effectively treats established lymphedema in preclinical mouse models[14,16,17]. These findings are supported by recent studies demonstrating that T cells potently inhibit lymphangiogenesis by producing anti-lymphangiogenic cytokines/growth factors, including interferon gamma (IFN-γ), interleukin (IL)-4, and transforming growth factor β-1 (TGF-β1; refs 18–21). Taken together, these findings suggest that infiltrating CD4$^+$ cells in lymphedematous tissues decrease lymphatic function through multiple mechanisms including induction of structural changes of lymphatic vessels secondary to tissue fibrosis and inhibition of collateral lymphatic vessel formation. Thus, strategies targeting generalized CD4$^+$ inflammatory responses or inhibition of Th2 differentiation may hold clinical promise for treating lymphedema. In addition, because lymphedema is primarily a disease of the skin and subcutaneous soft tissues of the extremities, it may be possible to use topical approaches to accomplish these goals, thereby avoiding systemic complications.

Tacrolimus is an anti-T-cell agent that is FDA-approved as a topical formulation and used to treat cutaneous inflammatory/fibrotic diseases including atopic dermatitis[22], psoriasis[23] and localized scleroderma[24]. Tacrolimus is a macrolide produced by the soil bacterium *Streptomyces tsukubaensis* that is well tolerated when used for prevention of transplant rejection and treatment of a variety of autoimmune diseases. It exerts its anti-T-cell properties by binding to FK-506-binding protein 12, thus inhibiting calcineurin, and ultimately decreasing IL-2 expression[25]. Because IL-2 is essential for T-cell activation and differentiation of CD4$^+$ T cells, calcineurin inhibitors have profound CD4$^+$ cell immunosuppressive effects[26,27].

Considering that CD4$^+$ T cells, and their cytokines, play a critical role in lymphedema pathology, the purpose of this study was to evaluate the efficacy of the immunosuppressive drug, tacrolimus, applied topically for the prevention and treatment of lymphedema. Using mouse tail lymphedema and popliteal lymph node dissection (PLND) models, we show that topical tacrolimus potently prevents development of lymphedema after lymphatic injury, as well as alleviates pathologic changes of established lymphedema. Topical tacrolimus has low systemic absorption, but markedly decreases skin CD4$^+$ cell infiltration and fibrosis while increasing formation of lymphatic collaterals. In addition, animals treated with tacrolimus had increased lymphatic collecting vessel-pumping with significantly improved lymphatic function. These findings have important implications for the treatment of lymphedema and represent a paradigm change for treatment since previous experimental attempts have focused exclusively on increasing lymphatic growth and function using exogenous lymphangiogenic growth factors.

## Results

**Tacrolimus decreases lymphedema without immunosuppression.** To study the effect of topical tacrolimus on lymphedema, we used a well-described mouse tail model of lymphedema[28–32]. Disruption of the superficial and deep lymphatics of the mouse tail resulted in a greater than 100% increase in tail volumes 2 weeks after surgery (Fig. 1a,b). We have previously shown that swelling at this time point is due primarily to accumulation of interstitial fluid[14]. Chronic lymphatic obstruction in the tail results in gradual replacement of interstitial fluid by fibroadipose tissue, as well as accumulation of inflammatory cells, occurs over the ensuing 4 weeks[14]. These pathologic changes closely mirror clinical lymphedema and persist for at least 6–9 additional weeks once lymphedema is established[28–32]. On the basis of this knowledge, we used two different tacrolimus treatment approaches. In one group of animals, we treated tacrolimus beginning 2 weeks after surgery to 6 weeks in an effort to prevent development of lymphedema (that is, early treatment). In another group we waited 6 weeks after surgery for lymphedema to become established and then treated with tacrolimus until 9 weeks with intent to treat these established soft tissue changes (that is, late treatment). For both the approaches we treated the animals with 0.1% tacrolimus or vehicle control (petroleum jelly), twice daily for 4 week period (for early treatment) and 3 week period (late treatment). Tacrolimus (~0.05 g) was applied as a thin layer to the entire tail area.

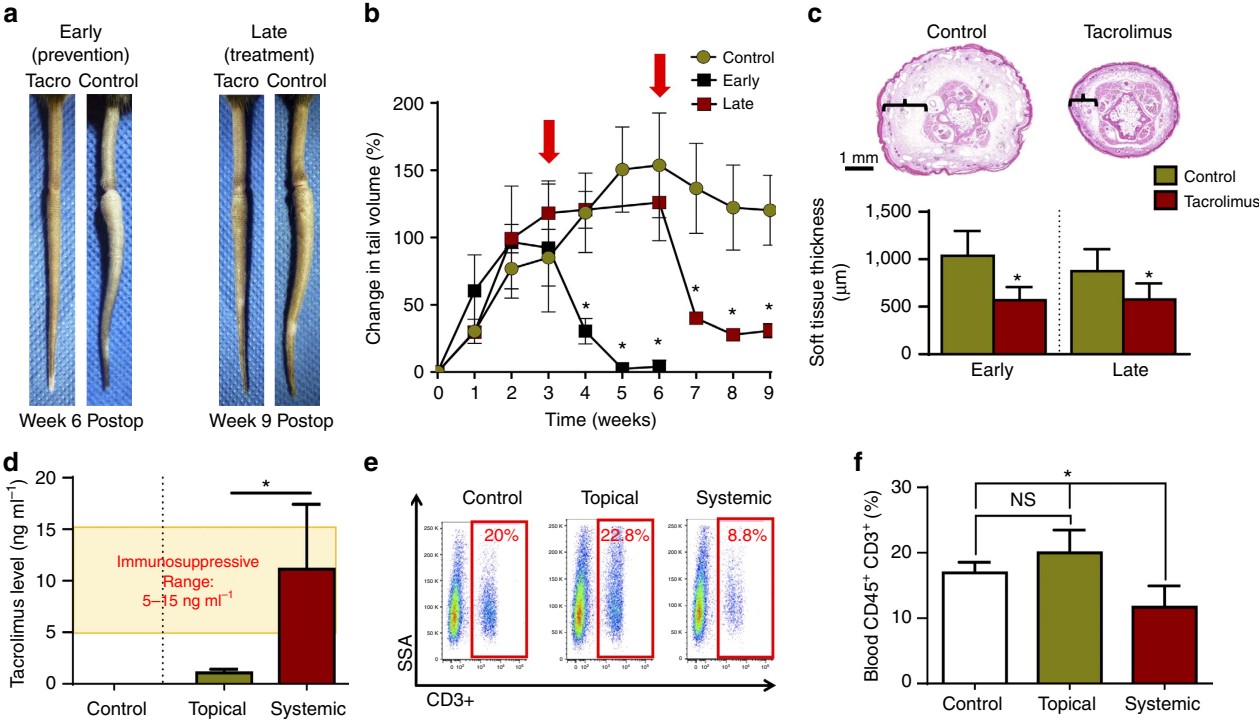

**Figure 1 | Topical tacrolimus decreases tail lymphedema.** (**a**) Representative photographs of mouse tails after surgical excision of superficial/deep collecting lymphatics and treatment with or without topical tacrolimus beginning either 2 weeks (early treatment) or 6 weeks (late treatment) after surgery. Images represent 4 weeks of treatment in the early group and 3 weeks of treatment in the late group. (**b**) Graphical representation of tail volume changes after early (*$P = 0.021$) or late (*$P = 0.018$) treatment with tacrolimus, as compared with controls (red arrows indicate timing of treatment) ($n = 8$/group). (**c**) Representative cross-sectional histological images (upper panel) of control and early tacrolimus-treated mouse tails harvested 6 weeks after lymphatic ablation. Brackets show soft tissue thickness. Quantification of soft tissue changes (lower panel) after early or late treatment with tacrolimus (*$P \leq 0.001$) ($n = 8$/group). (**d**) Quantification of whole-blood tacrolimus levels demonstrating immunosuppressive concentrations in systemically treated animals (4 mg kg$^{-1}$ intraperitoneally daily) and non-immunosuppressive levels in the topically treated group (*$P = 0.007$; $n = 6$/group). (**e**) Representative flow cytometry plots displaying side scatter area (SSA) on the $y$ axis and CD3$^{+}$ cells (T cells) on the $x$ axis for animals retreated with vehicle control, topical tacrolimus or systemic tacrolimus. (**f**) Quantification of blood T cells for each group. Note significant reduction in T cells only in systemically treated animals (*$P = 0.012$; $n = 6$/group). Experiments were repeated two to three times. All data represent mean ± s.d. with $P \leq 0.05$ considered as significant. Two-tailed Student's $t$-test (**c**,**d**) and analysis of variance (ANOVA) with *post hoc* tests (**b**,**f**).

Early treatment with topical tacrolimus markedly decreased tail swelling and prevented development of permanent swelling (Fig. 1a,b). Gross examination of the tails from experimental animals demonstrated a near complete resolution of lymphedema, and this change corresponded to a 95% decrease in tail volume and nearly 50% decrease in soft tissue thickness (Fig. 1b,c). Late treatment was also highly effective in decreasing gross tail swelling, tail volume and soft tissue thickness, as compared with controls, although in these animals the tail volumes did not return to preoperative levels (Fig. 1b,c). We also analysed systemic levels of tacrolimus and peripheral T-cell counts to determine whether topically applied tacrolimus is absorbed in an appreciable manner. This analysis demonstrated that systemic absorption of topical tacrolimus (mean value of 1.06 ng ml$^{-1}$) remained significantly below the therapeutic immunosuppressive levels achieved with systemic administration (5–15 ng ml$^{-1}$; Fig. 1d). In addition, animals treated with topical tacrolimus showed no changes in circulating blood T cells or CD4$^{+}$ cells, as compared with vehicle-treated controls (Fig. 1e,f and Supplementary Fig. 1).

**Tacrolimus decreases inflammation and fibrosis.** Chronic inflammation is a histological hallmark of clinical lymphedema and is characterized by increased accumulation of T-helper cells,

T regulatory cells and macrophages[14,16,33,34]. Consistent with this, we found that tacrolimus-treated animals had markedly decreased numbers of leukocytes infiltrating the dermis and subcutaneous fat as compared with controls (CD45$^{+}$ cells; 56% reduction-early treatment; 49% late treatment; Fig. 2a). Inflammatory cells in lymphedematous tissues harvested from control animals were located in close proximity to the capillary and collecting lymphatics, but were virtually absent in tacrolimus-treated mice. Similarly, we noted marked decreases in the numbers of infiltrating CD3$^{+}$ cells (53% reduction-early treatment; 49% late treatment; Supplementary Fig. 2A), CD4$^{+}$ cells (78% reduction-early treatment, 71% late treatment; Fig. 2b) and IFN-γ-producing cells (54% reduction-early treatment, 57% late treatment; Fig. 2c). In addition, we noted a decrease in the soft tissue infiltration of macrophages (F4/80$^{+}$ cells; 86% reduction-early treatment; 73% late treatment; Supplementary Fig. 2B). Taken together, these findings show that following lymphatic injury, inflammatory cells accumulate in large numbers in close proximity to skin/subcutaneous lymphatic vessels and that this response is mitigated by topical application of tacrolimus.

Patients with lymphedema have progressive soft tissue fibrosis and the degree of fibrosis correlates with the severity of disease[35]. Therefore, we analysed several markers of fibrosis in the tail tissues to understand the effects of tacrolimus treatment on

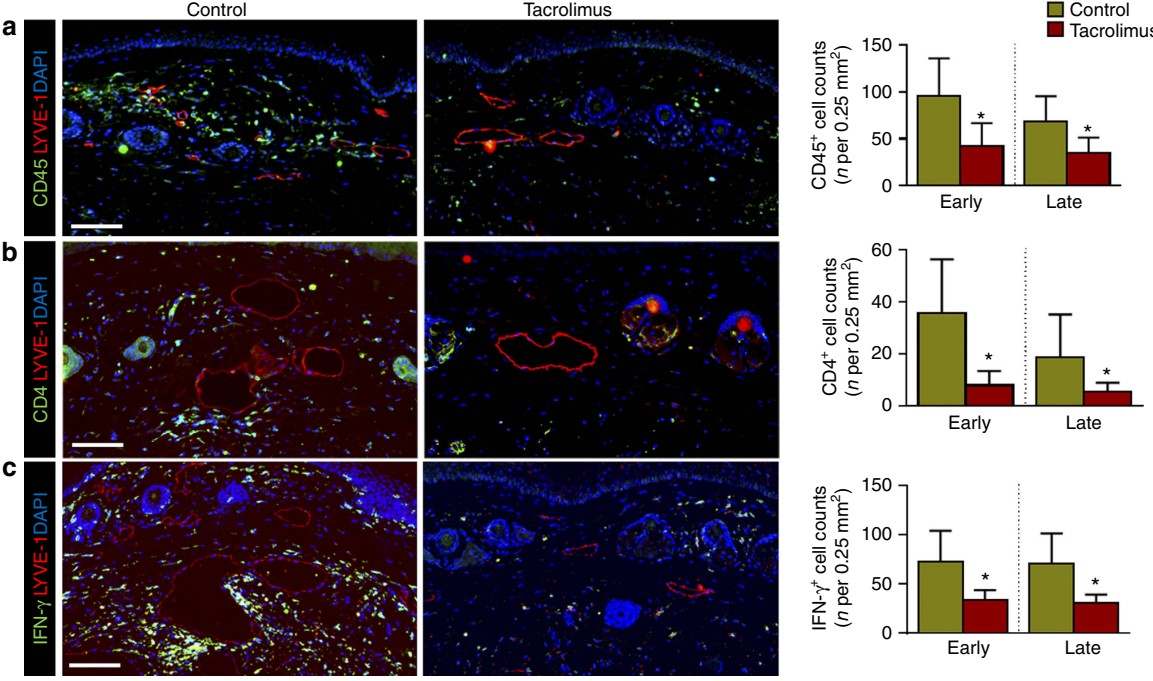

**Figure 2 | Topical tacrolimus decreases inflammation after lymphatic injury.** (a–c) Representative × 40 images of tail tissue sections from control and topical tacrolimus-treated animals 6 weeks after surgery with immunofluorescent localization of CD45$^+$(a), CD4$^+$ cells (b) and IFN-γ$^+$ cells (c). Lymphatic vessels are stained for LYVE-1$^+$ (red) in each figure. Quantifications of positive cell per 0.25 mm$^2$ area (four random areas per mouse) for both early and late treatments are shown to the right of each figure ($P<0.001$ for all; $n=8$/group). Experiments were repeated two to three times. All data represent mean ± s.d. with $P≤0.05$ considered as significant. Data analysed by two-tailed Student's $t$-test. Scale bars, 100 μm.

this aspect of the disease. We found that topical tacrolimus treatment markedly decreased dermal and subcutaneous type I collagen deposition and Scar index (picrosirius red birefringence measuring the ratio of collagen I/III), as compared with control mice (Fig. 3a,b). Lymphatic vessels of control mice were surrounded by thick layers of type I collagen; in contrast, tacrolimus-treated animals had essentially normal lymphatic vessels. Consistent with this observation and our previous reports[36], we also found that tacrolimus treatment markedly decreased expression of the pro-fibrotic growth factor TGF-β1 and cellular expression of its activated downstream signaling molecule, phosphorylated SMAD3 (pSMAD-3; Supplementary Fig. 3 and Fig. 3c). The degree of this response was similar for both early and late tacrolimus treatments.

**Tacrolimus increases lymphatic function**. To assess lymphatic function we performed near-infrared (NIR) lymphangiography with indocyanine green (ICG), which has been described as an effective means of quantifying lymphatic function in humans, pigs and mice by enabling real-time imaging of lymphatic vessels, calculation of packet frequency (or pulsatile flow of lymphatic fluid), and analysis of dermal backflow and dye clearance[37–39]. Using NIR imaging 6 weeks after lymphatic ligation, we noted rapid transport of interstitial fluid proximally across the tail wound in animals in which treatment was started 2 weeks after lymphatic injury (early treatment; Fig. 4a). In contrast, control animals demonstrated pooling of ICG distal to the lymphatic excision site with no transport across the zone of injury. This finding was confirmed using technetium-99 m ($^{99m}$Tc) lymphoscintigraphy, a technique in which a radiotracer is injected in the distal tail and uptake by the sacral lymph nodes is measured over time. Decay-adjusted uptake of the sacral lymph nodes in the early treatment animals

demonstrated a more than sixfold increase in $^{99m}$Tc uptake in tacrolimus-treated animals as compared with controls (Fig. 4b). Late treatment with tacrolimus similarly increased nodal uptake (twofold); however, this difference did not reach statistical significance.

Given the efficacy of tacrolimus in preventing and treating lymphedema in the tail model, we next sought to study how tacrolimus modulates lymphatic function using a more clinically relevant, PLND model (Supplementary Fig. 4)[40]. This model is relevant to the pathology of lymphedema since in Western countries typically the inciting event in this disease is usually lymphatic damage during the course of cancer treatment. Before assessing the effect of tacrolimus on lymphatic function, we first utilized the PLND mouse model to better understand the mechanisms by which chronic inflammatory reactions are activated after lymphatic injury. Previous studies have described dermal backflow as pooling of ICG into the interstitial space over time because of impaired lymphatic clearance and leaky lymphatic vessels[40–42]. Consistent with these studies, using NIR lymphangiography performed 4 weeks after PLND, we found that control animals had marked dermal backflow in initial lymphatics in the foot pad (Fig. 4c; indicated by white arrow) and decreased pumping capacity in collecting lymphatics of the hindlimb (Fig. 4d,e). More importantly, we found that these pathologic changes were markedly improved with topical tacrolimus as demonstrated by decreased dermal backflow and significantly increased packet transport of the collecting lymphatic vessel as compared with controls (more than twofold increase; Fig. 4c–e and Supplementary Movie 1). This response, similar to our findings with the tail model, correlated with a significant decrease in perilymphatic infiltration of inflammatory cells in mice treated with tacrolimus showing 39% reduction in number of CD45$^+$ cells (Fig. 4f upper panel, g) 56% reduction in CD4$^+$ cells (Supplementary Fig. 5) and

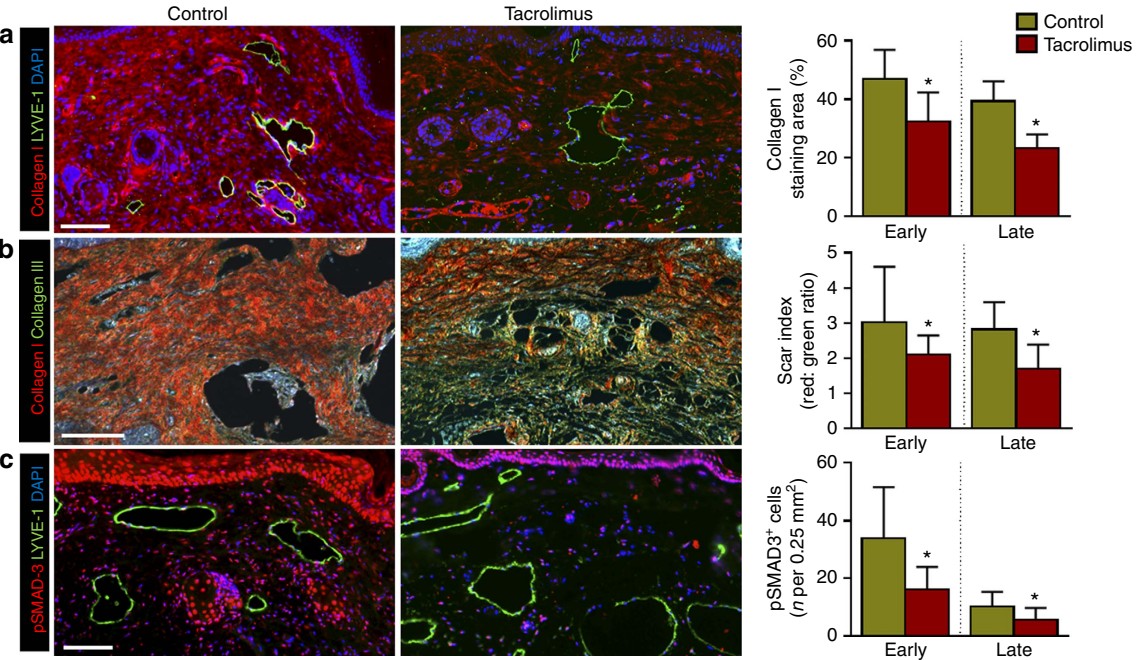

**Figure 3 | Topical tacrolimus decreases fibrosis in lymphedema. (a)** Representative × 40 images of tail tissues from control or early tacrolimus-treated mice 6 weeks after surgery with immunofluorescent localization of type I collagen (red) and lymphatic vessels (green). Quantifications of the percent area of collagen I staining in both early and late treatment are shown to the right (*P* < 0.001 for both; *n* = 8/group). **(b)** Representative × 40 images of picrosirius red staining of tail tissues harvested from control or early tacrolimus-treated animals (red-orange birefringence represents collagen I deposition; green-yellow birefringence represents collagen III deposition). Quantification of Scar Index (red: green ratio) is shown to the right (*P* = 0.036 early; *P* < 0.001 late; *n* = 8/group). **(c)** Representative × 40 immunofluorescent co-localization of pSMAD-3 (red) with lymphatic vessel (green) in tail tissues harvested from animals treated with control or early treatment with tacrolimus 6 weeks after surgery. Quantification of the number of positive cells per 0.25 mm² area (four random areas per mouse) for both is shown to the right of (*P* = 0.002 early, *P* = 0.026 late; *n* = 8/group). Experiments were repeated two to three times. All data represent mean ± s.d. with *P* ≤ 0.05 considered as significant. Data analysed by two-tailed Student's *t*-test. Scale bars, 100 μm.

36% reduction in F4/80$^+$ cells (Supplementary Fig. 6) compared with vehicle-treated controls.

In addition, treatment with tacrolimus resulted in a significant decrease in expression of inducible nitric oxide synthase (iNOS) by inflammatory cells (42% reduction in the number of iNOS-expressing cells; Fig. 4f lower panel, h). Importantly, changes in lymphatic contractility in response to topical tacrolimus treatment were only observed in the setting of lymphatic injury since treatment of non-operated animals (that is, anaesthesia only but no surgery) with tacrolimus did not result in changes in lymphatic contraction or inflammatory cell counts (Supplementary Fig. 7 and Supplementary Movie 2). In addition, to ensure that the observed effects of tacrolimus on lymphatic function were not a result of decreased vascular permeability and blood vessel leakage, we performed Miles assay to measure blood vessel permeability following PLND surgeries with and without tacrolimus. We observed no differences in blood vessel permeability between tacrolimus-treated and vehicle control. In addition, based on immunohistological staining for blood capillaries we observed no differences in blood vessel density between control and tacrolimus-treated lymphedema tail skins. Cumulatively, these data suggest that the effects of tacrolimus in increasing lymphatic function are indeed because of increased lymphatic function rather than decreased vascular density and leakage (Supplementary Fig. 8A–C).

Because structural changes in collecting lymphatic vessels (hypertrophy of smooth muscle, constriction of the lumen) have been described in human patients with lymphedema[8], we examined the collecting lymphatic vessels of our animals for similar changes. We found no differences in the amount of smooth muscle surrounding the vessels or in the luminal area of the vessels, suggesting that the observed tacrolimus-associated increase in lymphatic vessel contraction after PLND was not due to structural changes on the main collecting vessels (Supplementary Fig. 9A–D).

**Tacrolimus increases collateral lymphatic vessel formation.** Because T cells are known to potently inhibit lymph node lymphangiogenesis[18] and inflammatory lymphangiogenesis during wound repair[43], we next sought to determine whether treatment with tacrolimus increases the formation of collateral lymphatics. Indeed, histological analysis of tail wounds and identification of lymphatics using LYVE-1 immunofluorescent (IF) staining demonstrated a marked increase in newly formed lymphatic vessels bridging the zone of lymphatic injury (189% increase after early treatment; 106% increase after late treatment; Fig. 5a). Lymphangiographic analysis by NIR imaging, and IF staining for lymphatic vessels in hindlimbs 4 weeks following PLND, confirmed our tail model findings demonstrating that animals treated with tacrolimus consistently had significantly more collateral lymphatics draining towards the inguinal lymph node, thereby bypassing the zone of injury (Fig. 5b). In addition, we noted that number of proliferating lymphatic endothelial cells (LECs) (Ki67$^+$ by immunofluorescence) was increased in the tacrolimus-treated group (Supplementary Fig. 10A,B), but morphology of lymphatic vessels remained the same with no obvious vessel dilation or tortuosity. This lymphangiogenic response appeared to be independent of *Vegf-c* and *Vegf-a* expression since we noted no differences in

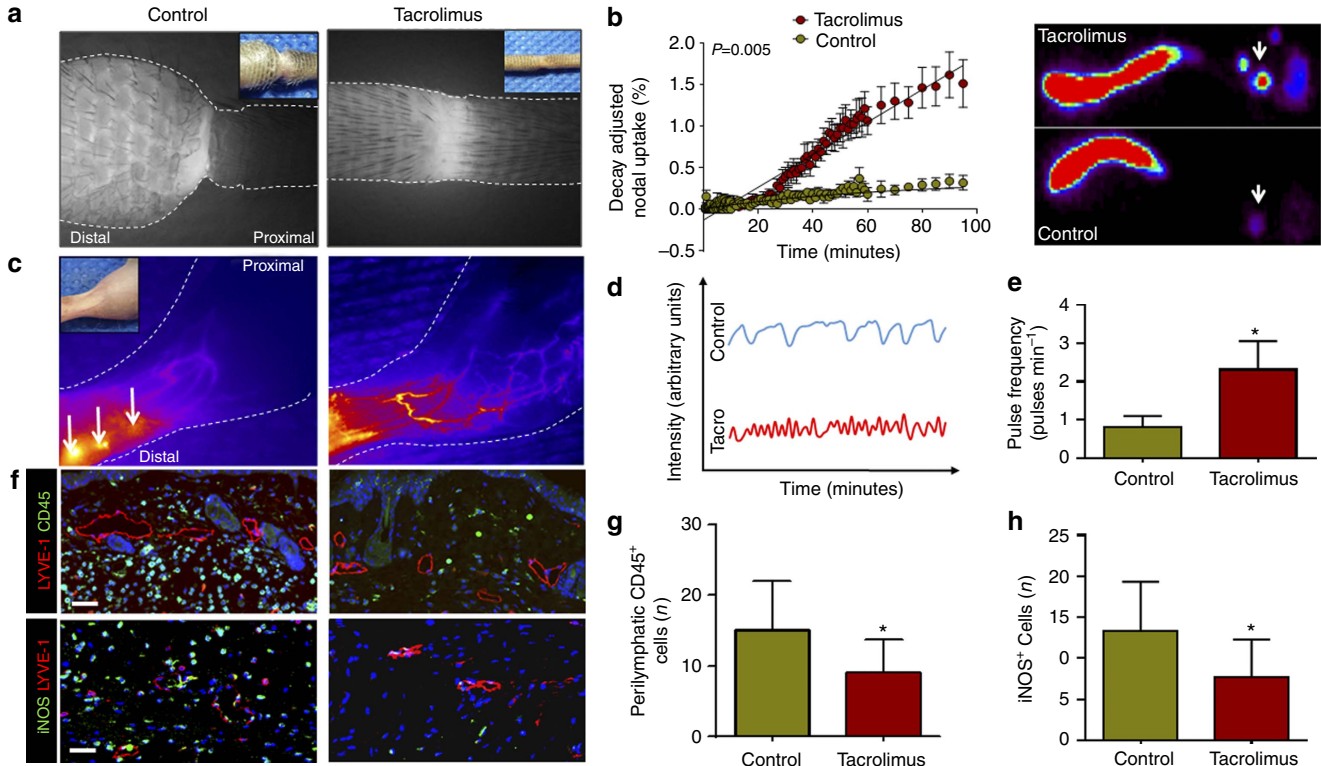

**Figure 4 | Tacrolimus improves lymphatic function after surgical lymphatic injury.** (**a**) Representative ICG images of mouse tails are shown 6 weeks after surgery following early treatment with or without tacrolimus. Note the flow of ICG proximally across the wound in tacrolimus-treated animals ($n = 6$/group). Inset shows photograph of same mice for orientation. (**b**) Decay-adjusted uptake of $^{99m}$Tc by sacral lymph nodes 6 weeks after surgical ligation of superficial/deep lymphatics in control and early tacrolimus (beginning 2 weeks after surgery)-treated animals (left panel; *$P = 0.005$; $n = 6$/group). Representative heat maps are shown in the right panel with the white arrow pointing towards the sacral lymph nodes ($n = 6$/group). (**c**) Representative ICG images of hindlimbs obtained 50 min after distal foot injection in mice treated with or without tacrolimus 4 weeks after PLND ($n = 6$/group). White arrows show dermal backflow (increased red colour) of ICG. Inset photograph is for orientation. (**d**,**e**) Graphical representation of lymphatic vessel pulsations in hindlimb collecting vessels of mice treated with or without tacrolimus 4 weeks after PLND. Quantification of pulsation frequency is shown to the right (*$P = 0.001$; $n = 6$/group). (**f**) Representative × 40 fluorescent co-localization images of inflammatory cells (CD45$^+$, green; upper panel), iNOS$^+$ cells (green; lower panel) and lymphatic vessels (LYVE-1$^+$, red) in tissues harvested from the distal hindlimbs of animals treated with control or tacrolimus 4 weeks after PLND. Note perilymphatic accumulation of CD45$^+$ and iNOS$^+$ cells. (**g**) Quantification of perilymphatic CD45$^+$ (**h**) and iNOS$^+$ cells in 0.25 mm$^2$ area (four random areas per mouse) of distal hindlimb tissues of control or tacrolimus-treated animals harvested 4 weeks after PLND (*$P = 0.0001$ (**g**) and *$P = 0.0007$ (**h**); $n = 6$). Experiments were repeated two to three times. All data represent mean ± s.d. with $P \leq 0.05$ considered as significant. Data analysed by two-tailed Student's $t$-test. Scale bars, 50 μm.

*Vegf-c* mRNA expression between control and tacrolimus-treated animals, as well as no difference between protein concentrations of VEGF-C and VEGF-A (Fig. 5c). However, consistent with our IF staining analysis, we noted a significant decrease in the expression, as well as protein concentrations, of four potently anti-lymphangiogenic growth factors and cytokines, TGF-β1 (refs 21,36), IFN-γ (refs 18,20), IL-4 and IL-13 (refs 19,44; Fig. 5c,d).

To determine the lymphangiogenic effects of tacrolimus in other inflammatory and wound models in which drainage of lymphatic fluid is not obstructed surgically, we next used two other models of inflammatory lymphangiogenesis. The cornea is a useful tissue for studying lymphangiogenesis because it is normally devoid of both blood and lymphatic vessels but develops both in the setting of inflammation[45]. We placed sutures in the corneas of mice and treated them with systemic tacrolimus or vehicle control daily for 2 weeks and found that tacrolimus treatment resulted in a significant increase in the proliferation of lymphatic (48% increase) but not blood vessels, as represented by an increase in the number of branch points (Fig. 6a–c). We also studied lymphangiogenesis during wound healing using an ear punch wound model and applying topical

tacrolimus or control ointment for 4 weeks. Similar to the corneal model, we found that topical tacrolimus significantly increased lymphatic vessel density and branching in the ear skin adjacent to the wound as compared with controls (Fig. 6d–f). More importantly, except for the increased density, sprouting filopodia and branching points tacrolimus-treated lymphatic vessels are morphologically very similar to controls in regards to tortuosity and vessel thickness (Supplementary Fig. 11A). To test the possibility of direct lymphangiogenic effects of tacrolimus, we applied tacrolimus to unwounded mouse ears for 4 weeks, and then performed whole-mount confocal imaging of lymphatic vessels. We observed no increase in lymphangiogenesis or lymphatic thickness in this uninjured, non-inflamed setting (Supplementary Fig. 11B). To corroborate the *in vivo* results, we have performed a series of *in vitro* experiments to test the direct effect of tacrolimus on LEC migration, tube formation and proliferation. Not surprisingly, we observed no significant differences in LEC migration, tube formation and proliferation between tacrolimus-treated (100 nM, 1 μM) and vehicle-treated LECs (Supplementary Fig. 12A–C). Together, these results indicate that tacrolimus facilitates the formation of new

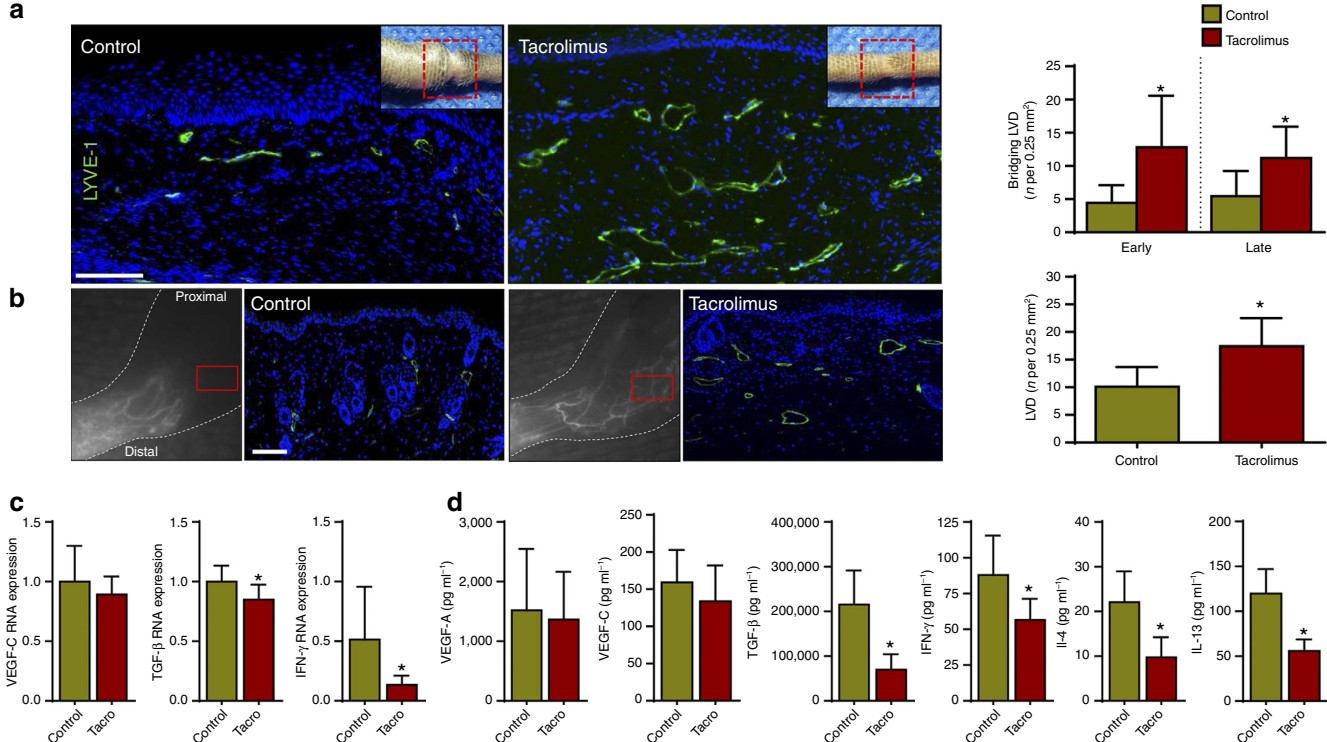

**Figure 5 | Topical tacrolimus increases formation of collateral lymphatic vessels.** (**a**) Left panel: representative longitudinal immunofluorescent × 40 images of LYVE-1 vessels (green) bridging the surgically created tail wounds of control and early-treated tacrolimus mice harvested 6 weeks after lymphatic injury; inset shows area where longitudinal sections were obtained. Right panel: quantification of bridging lymphatic vessel density (LVD) in the wounded portion of the tail in control versus early or late-treated tacrolimus mice ($P < 0.001$ for both; $n = 6$/group). (**b**) Left panel: representative ICG (left panels) and × 40 immunofluorescent images of LYVE-1$^+$ vessels (right panel showing area in red box) in control and tacrolimus-treated animals 4 weeks following PLND. Right panel: quantification of collateral lymphatic LVD in the anterolateral thigh region of animals treated with control or tacrolimus ($P < 0.001$; $n = 6$/group). (**c**) qPCR of RNA harvested from control and early-treated tacrolimus mouse tail tissues harvested 6 weeks after lymphatic injury, demonstrating relative expression of VEGF-C ($P = 0.264$), TGF-β1 ($P = 0.006$) and IFN-γ ($P = 0.022$; $n = 6$/group). (**d**) Protein concentrations from hindlimb tissues for each group using enzyme-linked immunosorbant assay for VEGF-A ($P = 0.711$), VEGF-C ($P = 0.233$), TGF-β1 ($P < 0.001$), IFN-γ ($P = 0.022$), IL-4 ($P = 0.034$) and IL-13 ($P = 0.009$; $n = 6$/group). Experiments were repeated two to three times. All data represent mean ± s.d. with $P \leq 0.05$ considered as significant. Data analysed by two-tailed student's $t$-test. Scale bars, 100 μm.

lymphatic vessels in the setting of inflammation generally and in the setting of lymphedema/lymphatic injury specifically.

## Discussion

In this study, using two different mouse models of lymphatic injury, we show that topical tacrolimus can prevent the development of lymphedema and is also efficacious for treating lymphedema once it is established. These findings are significant since there are currently no pharmacologic therapies available for the prevention or treatment of lymphedema. The clinically important concept here is that lymphedema may be treated with a local/topical application of an anti-inflammatory agent, thus decreasing the potential for systemic complications associated with this treatment. We found that tacrolimus was more effective when applied earlier after surgery, likely reflecting the fact that this treatment did not require reversal of established pathology. This finding is consistent with previous studies in other fibroproliferative disorders such as hepatic fibrosis in which prevention is much more effective than reversal of histological changes[46]. Since topical tacrolimus has limited systemic absorption, is currently FDA-approved for treatment of other dermatologic diseases[22] and has a long track record of safety, it may be possible for us to rapidly translate these findings to the clinical setting, thereby providing a novel therapeutic

option for this chronic and morbid disease. However, as with any translational treatment, additional studies will be required to determine the optimum concentration and delivery method for topical tacrolimus for the treatment of lymphedema. It is currently known that topically applied tacrolimus in its current formulation can penetrate to the level of the dermis[47]; however, additional studies and clinical trials will be needed to optimize this treatment for patients with lymphedema recognizing that the drug will be applied to a large surface area.

Our lab has shown the importance of CD4$^+$ cells and their cytokines in the pathology of lymphedema, as well as the development of fibrosis observed with lymphatic injury. For example, we have previously shown that athymic nude mice, mice deficient in CD4$^+$ cells or mice depleted of CD4$^+$ cells with neutralizing antibodies do not develop lymphedema or tissue fibrosis[14,16,48]. This response is specific for CD4$^+$ cells, since we have previously shown that depletion of CD8$^+$ cells or macrophages does not prevent fibrosis or lymphatic dysfunction. The findings of our current study are consistent with these previous reports, since we found that topically applied tacrolimus potently decreased inflammation and was highly effective in preventing fibrosis and lymphatic dysfunction following lymphatic injury. These findings are important because they represent a novel, clinically translatable intervention for the treatment of lymphedema. This conclusion is supported by

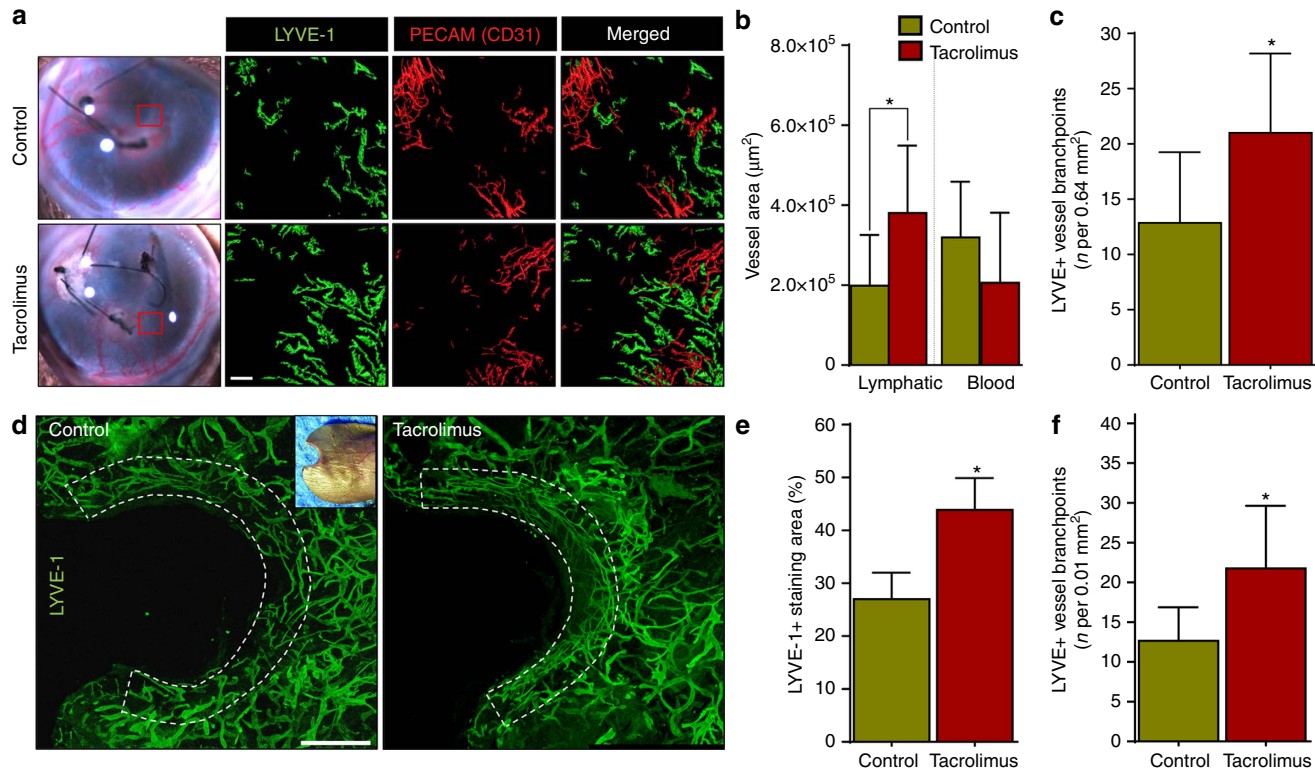

**Figure 6 | Tacrolimus increases inflammatory lymphangiogenesis. (a)** Representative gross (left panels) and whole-mount ×5 images (right panels) of mouse corneas stained for lymphatic vessels (High LYVE-1$^+$/low CD31$^+$, green) and blood vessels (CD31$^+$/LYVE-1$^-$, red) harvested 2 weeks after suture placement and treatment either with vehicle control or systemic tacrolimus. Red box denotes area shown in whole-mount images. Scale bar, 100 μm. **(b)** Quantification of corneal lymphatic (LYVE-1$^+$) and blood (CD31$^+$/LYVE-1$^-$) vessels (*$P = 0.032$, $n = 6$/group). **(c)** Quantification of lymphatic vessel branch points per 0.64 mm$^2$ area (four random areas/mouse) in suture placed cornea treated with or without tacrolimus demonstrating increased branching in tacrolimus-treated animals ($P = 0.033$; $n = 6$/group). **(d)** Representative immunofluorescent whole-mount ×5 images of ear wounds localizing LYVE-1$^+$ in control or topical tacrolimus-treated animals harvested 4 weeks after wounding. Inset photograph shows area where sections were obtained. Scale bar, 800 μm. **(e)** Quantification of the LYVE-1$^+$ staining area in ear skin (within 400 μm of the wound) demonstrating an increase in lymphangiogenesis in tacrolimus-treated animals ($P < 0.001$) ($n = 6$/group). **(f)** Quantification of lymphatic vessel branch points per unit area in ear wounds treated with or without tacrolimus, demonstrating increased branching in tacrolimus-treated animals ($P < 0.001$; $n = 6$/group). Experiments were repeated two to three times. All data represent mean ± s.d. with $P \leq 0.05$ considered as significant. Data analysed by two-tailed Student's t-test.

other clinical studies demonstrating the utility of topical tacrolimus in the treatment of fibrotic skin disorders including atopic dermatitis[22], keloids[49] and localized scleroderma[24,50,51].

We also noted that treatment with tacrolimus significantly increased lymphangiogenesis, lymphatic vessel branching and formation of collateral lymphatics after lymphatic injury and in response to inflammation. These findings are important because lymphangiogenesis and formation of collateral vessels provide a pathway to reroute lymphatic fluid away from the zone of lymphatic injury leading to resolution of lymphatic stasis[52]. Although regulation of lymphangiogenesis is complex, our studies suggest that increased formation of collateral lymphatics secondary to tacrolimus therapy is not secondary to increased expression of lymphangiogenic cytokines such as VEGF-C or VEGF-A. Instead, our results suggest that lymphangiogenesis in this setting is regulated by decreased anti-lymphangiogenic mechanisms including T-cell inflammatory responses and expression of anti-lymphangiogenic growth factors including TGF-β1, IFN-γ, IL-4 and IL-13. This hypothesis is supported by previous studies demonstrating profound anti-lymphangiogenic roles for T cells during inflammatory lymphangiogenesis[18–20], as well as studies demonstrating direct anti-lymphangiogenic effects by TGF-β1 (refs 21,36) and IFN-γ (ref. 20) even in a microenvironment

with abundant pro-lymphangiogenic growth factors, such as VEGF-C[28,43]. Taken together, these studies suggest that the balance between pro- and anti-lymphangiogenic forces play a key role in lymphatic regeneration after injury and that this balance is shifted towards lymphatic repair by application of topical tacrolimus. Future studies, however, will be required to determine temporal and spatial changes in growth factor expression and cellular infiltration to definitively answer these questions.

An interesting finding of our study was that topical tacrolimus significantly increased collecting vessel-pumping frequency after PLND. This finding is consistent with previous clinical studies demonstrating that patients with lymphedema have significantly decreased lymphatic pumping frequency and amplitude as compared with normal controls[53]. Although the precise mechanisms regulating this response remain unknown, this finding may be related to decreased perilymphatic inflammation and expression of iNOS. This hypothesis is supported by previous studies, demonstrating that the expression of iNOS by infiltrating inflammatory cells disrupt endogenous gradients of endothelial derived nitric oxide and impair lymphatic relaxation/pumping in response to cutaneous inflammatory responses[54–56]. This hypothesis is also supported by our finding that tacrolimus treatment of

animals without lymphatic injury (and therefore no perilymphatic inflammation) had no effect on lymphatic pumping frequency. An alternative hypothesis could be that lymphatic disruption and ligation decreases lymphatic pumping as a result of increased afterload in distally obstructed collecting lymphatics, and that this effect may be diminished in tacrolimus-treated animals because of augmented formation of collateral vessels. This hypothesis is supported by previous studies using isolated lymphatic vessels, demonstrating that changes in preload and afterload have significant effects on lymphatic pumping function[57,58]. Alternatively, decreased pumping in lymphatic collectors after lymphatic injury may reflect decreased preload resulting from impaired function or number of initial lymphatics secondary to increased perilymphatic collagen deposition, impaired lymphangiogenesis or capillary lymphatic leakiness. Most likely, changes in lymphatic pumping are multifactorial and will require additional study in the future. However, our results are encouraging since they show that a simple topical intervention can effectively increase lymphatic function.

In conclusion, we show that topical tacrolimus has low systemic absorption but potently decreases dermal and subcutaneous T-cell infiltration and tissue fibrosis after lymphatic injury. These changes prevent development of lymphedema and can reverse pathologic changes once lymphedema is established. Treatment with tacrolimus increases lymphatic function by increasing formation of collateral lymphatics and by increasing collecting lymphatic pumping frequency. To our knowledge, this is the first targeted topical pharmacologic means of preventing and treating post-surgical lymphedema.

## Methods

**Animal models and treatments.** The studies were performed using adult female (10–14 week old) C57BL/6 J mice (Jackson Laboratories, Bar Harbor, ME) that were maintained in light- and temperature-controlled pathogen-free environments and fed *ad libitum*. All studies were approved by the Institutional Animal Care and Use Committee (IACUC) at Memorial Sloan Kettering Cancer Center. Each experiment was performed using a minimum of six to eight animals, and assays were performed in triplicate. All cell counts were performed by reviewers blinded to the intervention.

Tail surgery and lymphatic ablation were performed as previously published[12]. Briefly, the superficial and deep collecting lymphatics of the mid portion of the tail were excised using a 2-mm circumferential excision. Tacrolimus 0.1% (Astellas, Tokyo, Japan)/vehicle control (petroleum jelly) was treated topically beginning at 2 weeks after surgery (early treatment) and beginning at 6 weeks after surgery (late treatment) in different set of animals. For both approaches we treated the animals with 0.1% tacrolimus or vehicle control, twice daily for a 4-week period (for early treatment) or a 3-week period (late treatment). Tacrolimus (∼0.05 g) was applied as a thin layer to the entire tail area. To enable analysis of the lymphatic collecting vessel-pumping capacity, we utilized a previously described mouse popliteal lymphadenectomy model[40,59]. Briefly, the hindlimb collecting vessels and popliteal lymph nodes were identified, and the lymph nodes were excised with the popliteal fat pad. Beginning 2 weeks after surgery, animals were randomized to treatment with either 0.1% topical tacrolimus or vehicle control (petroleum jelly) twice daily for 2 weeks.

A corneal lymphangiogenesis assay was performed as previously reported[60]. Briefly, 10-0 nylon sutures (Ethicon, Cincinnati, OH) were placed in the cornea at 120° angles. Beginning immediately after suture placement, animals were treated with systemic tacrolimus or vehicle control daily for 2 weeks, followed by analysis using confocal microscopy (Leica Microsystems, Weitziar, Germany). Systemic tacrolimus (Biotang Inc., Lexington, MA) was dissolved in 10% ethanol with 1% tween 80 in PBS[61,62] and dosed at 4 mg kg$^{-1}$ intraperitoneally daily. Vehicle control for systemic tacrolimus was the equivalent volume of 10% ethanol, 1% tween 80 in PBS. Cutaneous lymphangiogenesis was assessed using an ear punch wound model as previously reported[63]. Following wounding, ear skin was treated either with topical tacrolimus or vehicle control for 4 weeks. Ears were then harvested and fixed in 1% paraformaldehyde (PFA) overnight. The anterior and posterior portions of the ear skin were then divided, removing the cartilage and whole-mount staining for LYVE-1 and CD31 was performed. Tile-scan images were obtained using a confocal microscope (Leica Microsystems) and skin within 400 μm of the wound edge was analysed using the Metamorph software (Molecular Devices, Sunnyvale, CA). Standardized (200 μm × 200 μm) fields were analysed for the number of branch points present per unit area by two blinded reviewers.

**Flow cytometry.** Flow cytometry was performed on peripheral blood samples as previously reported[16]. Briefly, erythrocytes were lysed with RBC lysis buffer (eBioscience, San Diego, CA) followed by staining with fluorophore-conjugated antibodies CD45 (Cat #103106; 1:100) CD3 (Cat #100236; 1:100) and CD4 (Cat #100406; 1:100; all from Biolegend, San Diego, CA) and analysis with a FACSCalibur flow cytometer (BD Biosciences, Franklin Lakes, NJ) using the FlowJo software (Tree Star, Ashland, OR).

**Tacrolimus blood level by mass spectrometry.** Blood levels of tacrolimus were measured using mass spectrometry in a modification of a previously reported method[64]. Briefly, whole blood was collected in sodium EDTA-coated tubes (Terumo, Shibuya, Japan) and then analysed using a Thermo Scientific Aria TLX-2 turbulent flow chromatograph coupled to a TSQ Quantum Ultra triple quadrupole mass spectrometer (Thermo Scientific, Franklin, MA). The TurboFlow column used was a Cyclone P-50 × 0.5 mm, while the analytical column was a Hypersil Gold C-18 column, 3 × 50 mm. To whole blood (50 μl) was added 200 μl of 0.2 mM $ZnSO_4$ containing ascomycin as an internal standard. Following a 30-min incubation and centrifugation, 50 μl of the supernatant was injected into the turbulent flow chromatograph system. The analytes were eluted through the column (0.75 ml min$^{-1}$) with a gradient of water and methanol solutions containing 10 mM ammonium formate and 0.1% formic acid. The analytical run time was 4.5 min. The between-day imprecision of the assay was determined at three concentrations over a period of 10 days. At concentrations of 3.3, 12.6 and 31.9 ng ml$^{-1}$ the coefficients of variation were of 9.8%, 7.0% and 7.8%, respectively. The assay has a linear range from 0 to 40 ng ml$^{-1}$.

**Analysis of lymphatic function.** Tail volumes were calculated using the truncated cone formula as previously reported[65] and confirmed using histological measurements of soft tissue thickness of the skin/subcutaneous tissues in a standardized manner using the Mirax Imaging Software (Carl Zeiss, Munich, Germany).

Lymphoscintigraphy was performed using our previously published methods[66]. Briefly, 50 μl of filtered technetium-99 m ($^{99m}$Tc) sulfur colloid (Nuclear Diagnostic Products, Rockaway, NJ) was injected in the distal tail. Images were taken using an X-SPECT camera (Gamma Medica, Northridge, CA) and region-of-interest analysis was performed to derive decay-adjusted counts in the sacral lymph nodes and to calculate peak and rate of nodal uptake using ASIPro software (CTI Molecular Imaging, Knoxville, TN).

NIR was performed using a modification of previously published results[35]. Briefly, 15 μl (0.15 mg ml$^{-1}$) ICG (Sigma-Aldrich, Saint Louis, MO) was injected intradermally in the web space of the dorsal hindlimb and visualized using an EVOS EMCCD camera (Life Technologies, Carlsbad, CA) with a LED light source (CoolLED, Andover, United Kingdom). Static/video images were obtained using a Zeiss V12 Stereolumar microscope (Caliper Life Sciences, Hopington, MA) and lymphatic pumping function was analysed using Fiji software (NIH, Bethesda, MD) by identifying a region-of-interest over the dominant collecting vessel of the leg and subtracting the background fluorescent intensity plotted over time.

**Histology and immunostaining.** Immunohistochemical staining was performed using our published methods[65]. Briefly, tissues were fixed in 4% PFA at 4 °C, decalcified in 5% sodium EDTA (Santa Cruz Biotechnology, Dallas, TX), embedded in paraffin and sectioned at 5 μm. Cut sections were rehydrated and heat-mediated antigen unmasking was performed using 90 °C sodium citrate (Sigma-Aldrich). Nonspecific binding was blocked with 2% BSA/20% animal serum. Tissues were incubated overnight with primary antibody at 4 °C. Primary antibodies used for immunohistochemical stains included goat anti-mouse LYVE-1(AF-2125; 1:300), rat anti-mouse CD45 (MAB-114; 1:300) and rat anti-mouse CD4 (BAM-554; 1:200; all from R&D, Minneapolis, MN). Rabbit anti-human/mouse CD3 (A0452; 1:300; Dako, Carpinteria, CA) and Cy3-conjugated mouse anti-αSMA (C6-198; 1:1,000; Sigma-Aldrich). Rat anti-mouse F4/80 (ab16911; 1:200), rabbit anti-human IFN-γ (ab9657; 1:200), rabbit anti-mouse TGF-β1(ab66043; 1:200), rabbit anti-mouse p-SMAD3 (ab51451; 1:200), rabbit anti-mouse collagen I (ab34710; 1:200), rabbit anti-mouse iNOS (ab3523; 1:200), rabbit-anti-mouse/human Ki67 (ab16667; 1:200), rabbit anti-human VEGFR-3 (ab27278; 1:200) and hamster-anti-mouse podoplanin (ab11936; 1:200; all from Abcam, Cambridge, MA). Rat anti-mouse CD31 (553370; 1:300; BD Biosciences).

IF staining was performed using AlexaFluor fluorophore-conjugated secondary antibodies (Life Technologies, Norwalk, CT). Images were scanned using Mirax imaging software (Carl Zeiss). Perilymphatic CD45$^+$ and CD4$^+$ cell counts were assessed by counting positively stained cells within 50 μm of the most inflamed lymphatic vessel in each quadrant of the leg. Positively stained cells were counted by two blinded reviewers in four randomly selected, × 40 high-power fields in a minimum of four fields per animal. Collagen I deposition was quantified using the Metamorph software (Molecular Devices) in dermal areas of 5 μm cross-sections. This analysis was confirmed using picrosirius red staining and scar index calculation as previously reported[67]. Bridging lymphatic vessels in

mouse tails were counted in the re-epithelialized surgical site in four different high-power fields per tail.

**Sirius red staining.** Paraffin sections of tail tissues were stained with the Picrosirius Red Staining Kit (Polysciences, Warrington, PA) according to the manufacturer's instructions. Images were obtained through polarized light on an Axiocam 2 microscope (Carl Zeiss), and the scar index was quantified with the Metamorph software by calculating the ratio of red-orange: green-yellow fibres with higher numbers representing increased scarring.

**Real-time PCR.** RNA extraction was performed on tail skin using TRIZOL (Invitrogen, Life Technologies) according the manufacturer's recommendations. RNA quality and quantity was assessed using an Agilent Bio Analyser (Agilent Technologies, Inc., Santa Clara, CA). The isolated RNA was converted to cDNA using a TaqMan reverse transcriptase kit (Roche, Branchburg, NJ) and relative expression of gene expression between groups was performed using delta–delta CT PCR analysis and normalizing gene expression using GAPDH RNA amplification as previously described[68]. Relative expression was calculated using the formula: 2[ − (Ct gene of interest − Ct endogenous control) sample A − (Ct gene of interest − Ct endogenous control) sample B)]. All samples were performed in triplicate. The primers used for the PCR targets of interest were for VEGF-C (Mm00437310_m1), TGF-β1 ( Mm01178820_m1) and IFN-γ (Mm01168134; Applied Biosystems, Life Technologies).

**Inflammatory cytokines and growth factor analysis.** Inflammatory cytokine analysis was performed on isolated hindlimb protein harvested from mice that underwent PLND for each group. Briefly, hindlimb skin tissue was harvested from mice, ground using a mortar and pestle following flash-freezing using liquid nitrogen, following which protein was extracted using a 100:1:1 mixture of tissue extraction protein reagent (Thermo Fisher Scientific, Waltham, MA), protease inhibitor and phosphatase inhibitor (Sigma-Aldrich). Following extraction, samples were spun using a centrifuge at 16,060g for 10 min and the supernatant was collected as our protein sample. Sample protein concentration was determined using the Bradford protocol for protein concentration. Protein concentrations were standardized and enzyme-linked immunosorbant assay was performed to measure inflammatory cytokines TGF-β1, INF-γ, IL-4 and IL-13 (eBioscience) and growth factors VEGF-A and VEGF-C (US Biological, Salem, MA) as per the manufacturer's recommendation, and all experiments were run in duplicate to ensure internal consistency.

**Miles assay for vascular permeability.** Miles assay for vascular permeability was performed as described[69]. Briefly, 200 μl of 0.5% sterile Evans blue was injected via tail vein to 2-week tacrolimus-treated PLND mice. After 30 min PLND hindlimbs were imaged to observe Evans blue leakage. Hindlimbs were excised and incubated in formamide for 48 h at 55 °C to extract the Evans blue. Extracted Evans blue was quantified by measuring absorbance at 610 nm.

***In vitro* LEC assays.** *In vitro* migration, tube formation and proliferation assays were performed as described[44]. Mycoplasma-free Human dermal LECs (PromoCell, Heidelberg, Germany) were used for all *in vitro* assays. Briefly, for migration assay confluent LEC monolayer in a six-well dish was scratched with 200 ml pipette tip and supplemented with 100 nM and 1 μM concentrations of tacrolimus and cultured for 24 h. Phase contrast images were acquired at 12 and 24 h, and wound closure area was measured and plotted to represent LEC migration. For tube formation, $1 \times 10^5$ LECs suspended in 300 μl of growth factor reduced matrigel (BD Biosciences) and plated in each well of a 24-well dish. Plates were cultured with EGM-MV2 media (PromoCell) supplemented with 100 nM and 1 μM concentrations of tacrolimus. At 24 h, phase contrast images of tubules were taken and analysed for tube length. For proliferation assay, LECs were cultured in chamber slides (Nunc, Rochester, NY) with 100 nM and 1 μM concentrations of tacrolimus. After 24 h, cells were fixed and stained for Ki67 (Abcam) and quantified based on confocal image analysis. In another set of experiments after 24 h culture of LECs with desired tacrolimus concentrations, EdU/5-ethynyl-2′-deoxyuridine (Life Technologies, Grand Island, NY) was added in culture media for a brief period. Cells were harvested, fixed, permeabilized and stained, and EdU-positive cells were quantified using Flow cytometry (LSRII; BC Biosciences, San Jose, CA). All the experiments were performed in quadruplicates or more, and DMSO was used as vehicle control.

**Statistical analysis.** A sample size of six to eight animals per group was chosen based on our previous experience with these models and preliminary data. Data were analysed and displayed using the GraphPad Prism software (GraphPad Software, La Jolla, CA). Values are presented as mean ± s.d. unless otherwise noted. Statistical significance was set at $P \leq 0.05$, and differences between the two groups was assessed with the two-tailed Student's *t*-test while multiple analyses were performed using analysis of variance with *post hoc* tests to compare within groups.

**Data availability.** All the relevant data are available from authors upon request.

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

## Acknowledgements

We thank Mesruh Turkekul for expert technical and histological assistance. We also thank the Molecular Cytology Core at MSKCC for assistance with both histology and tissue imaging. NIH R01 HL111130-01 awarded to B.J.M; Plastic Surgery Foundation Pilot Research Grant 312436 awarded to J.C.G.; Plastic Surgery Education Foundation Research Fellowship Grant 312441 awarded to J.C.G; NIH T32 CA009685-21A1 to G.G.N.; Memorial Sloan Kettering Summer Research Fellowship Grant 5R25CA020449 to D.K.J.; NIH/NCI Cancer Center Support Grant P30 CA008748.

## Author contributions

J.C.G., R.P.K. and B.J.M. conceived the concept and designed the experiments. J.C.G., R.P.K., I.L.S. and D.K.J. developed methods. J.C.G., R.P.K., G.E.H., J.S.T., I.L.S., G.D.G.N., M.D.N., R.C.S. and D.C.C. performed the experiments. J.C.G., R.P.K., G.E.H., J.S.T., I.L.S., G.D.G.N., M.D.N., R.P.K. and B.J.M. analysed the data. J.C.G., R.P.K., G.E.H., I.L.S., J.S.T., M.D.N., R.C.S., D.C.C. and B.J.M. prepared and edited the manuscript.

## Additional information

**Competing financial interests:** The authors declare no competing financial interests.

