## [Peer Review File · Nature Communications]

Reviewers' comments:

Reviewer #1 (expert in T cells and lymphangiogenesis)

Remarks to the Author:

The rationale of the manuscript by Gardenier et al is based on somewhat conflicting previous observations by various authors of the current manuscript regarding the role of CD4 T cell subsets in lymphangiogenesis. A recent paper by Savetsky et al in Plos one claims roles for IL4 and IL13 (TH2), whereas Katura et al claim IFN γ . The current manuscript claims TH2 cells are the crucial ones, and uses a calcineurin inhibitor (tacrolimus) (inhibits NFAT) to deplete T cells and inflammation and restore lymphatic vessel function. Several models are used, including tail wounding, lymph node removal, corneal graft suturing and ear wounding. In most cases, tacrolimus is administered topically in petroleum jelly, although in some cases it is administered subcutaneously. Treatment with tacrolimus results in quantifiable changes, including reduced local T cell counts, reduced IFN γ , reduced macrophages, reduced swelling, and enhanced lymphatic vessel function. These empirical observations suggest that inhibiting inflammation could be beneficial in some situations of acute tissue damage, but do not dig deep enough to reveal mechanisms.

1. Although the data taken together support the conclusion that locally applied tacrolimus could be beneficial in models of lymphedema, the studies are somewhat superficial. Instead of carrying out the same experiments on several models, a more in depth analysis of mechanism of one or two models would have been more informative.
2. The authors' premise is that TH2 cytokines inhibit lymphangiogenesis and they should refer to the paper by Savetsky et al (2015) that stresses the anti-lymphangiogenic roles of IL4 and IL13. However, they do not measure TH2 cytokines after tacrolimus; thus the mechanism is still unclear. Why are IL4 and IL13 not analyzed in light of the authors' previous data?
3. The authors study several different models of inflammation. In general, inflammation results in lymphangiogenesis, and even more lymphangiogenesis appears to be beneficial at least in the tail wounding model. How about in the cornea model (Figure 6?). Is the florid lymphangiogenesis induced by tacrolimus beneficial or harmful?
4. What is the mechanism of lymphangiogenesis that is stimulated by tacrolimus? Is this proliferation of existing vessels, sprouting, generation of new vessels? Does this differ by model? The authors claim that tacrolimus results in lymphangiogenesis. However, in many cases, there do not seem to be more lymphatic vessels in the treated mice (eg. Fig 3A, 3C and 3D). Although the authors nicely show in figure 4 an enhancement of lymphatic vessel function (increased pulsing frequency, increased lymph node dye uptake) after treatment and increased collateral vessel density in the LN removal model after tacrolimus. More quantification of lymphangiogenesis (e.g. by proliferation by e.g. Ki67) would be appropriate.
5. In supplemental Figure 10, the authors quantify LYVE-1+ staining area in controls and tacrolimus treated unwounded mouse ears. They need to carry out the same analysis in wounded mouse ears treated or not with tacrolimus.
6. Why does tacrolimus result in a decrease in lymphatic vessel pulsing intensity (Figure 4Di)?

Reviewer #2 (expert in lymphangiogenesis and lymphedema)

Remarks to the Author:

This is an elegant study illustrating the pro-lymphangiogenic effect of the T-cell inhibitor tacrolimus. The following issues should be considered.

Remarks

The authors should more consistently quantify other structural features of lymphangiogenesis besides lymphatic vessel density, such as lymphatic area, branching, morphology, etc in each model studied to support their conclusion that tacrolimus stimulates lymphangiogenesis.

An important control is missing: does tacrolimus affect (stimulate) lymphatic endothelial cells? A direct effect, independent of T cells, should be excluded.

Tacrolimus has beneficial effects in the models used, but these are surgical models, in which inflammation is induced due to the surgery. Does tacrolimus also work in other lymphedema models?

The link between PLND and ROS is unclear and insufficiently characterized. The authors also did not show that "lymphatic injury results in generation of ROS" - this is an overstatement.

Does tacrolimus affect blood vessels and their growth?

The authors should support their conclusions by treating T cells in vitro with tacrolimus or control, and testing whether the conditioned medium affects lymphatic endothelial cells. Pro-lymphangiogenic cytokine activity in the conditioned medium should then be immuno-neutralized for the mentioned cytokines.

Reviewer #1 (expert in T cells and lymphangiogenesis)

Remarks to the Author:

The rationale of the manuscript by Gardenier et al is based on somewhat conflicting previous observations by various authors of the current manuscript regarding the role of CD4 T cell subsets in lymphangiogenesis. A recent paper by Savetsky et al in Plos one claims roles for IL4 and IL13 (TH2), whereas Katura et al claim IFN γ . The current manuscript claims TH2 cells are the crucial ones, and uses a calcineurin inhibitor (tacrolimus)(inhibits NFAT) to deplete T cells and inflammation and restore lymphatic vessel function. Several models are used, including tail wounding, lymph node removal, corneal graft suturing and ear wounding. In most cases, tacrolimus is administered topically in petroleum jelly, although in some cases it is administered subcutaneously. Treatment with tacrolimus results in quantifiable changes, including reduced local T cell counts, reduced IFN γ , reduced macrophages, reduced swelling, and enhanced lymphatic vessel function. These empirical observations suggest that inhibiting inflammation could be beneficial in some situations of acute tissue damage, but do not dig deep enough to reveal mechanisms.

The rationale for our study is derived from several recent studies by our lab and others demonstrating that T cells have an anti-lymphangiogenic and profibrotic function in wound healing and after lymphatic injury. Our previous papers, which the reviewer nicely cites, clearly provide evidence that Th2 cytokines are potently anti-lymphangiogenic and promote tissue fibrosis after lymphatic injury. Similarly, Kataru *et al.* have shown that IFN- γ , a Th1 cytokine, also inhibits lymphangiogenesis by decreasing LEC proliferation, migration, and differentiation. The findings of these studies are not contradictory as the reviewer infers but rather complementary and strongly suggest that T cells can play an anti-lymphangiogenic role. Moreover, given that a major clinical component of lymphedema is fibrosis, these previous studies provide a rationale for using anti-fibrotic approaches aimed at T cells. By using tacrolimus topically in this study (not subcutaneously), we show that inhibiting T cell inflammatory responses can improve lymphangiogenesis and lymphatic function. We show that T cells are necessary for tissue fibrosis following lymphatic injury and that inhibition of this response with an FDA approved medication is highly effective in treating lymphedema in preclinical models. These findings are important and, contrary to the reviewer's position, dig deep into the pathology of lymphedema. More importantly, our study identifies a therapeutic approach in a disease that is currently only treated with palliative treatments such as compression and physical therapy.

1. Although the data taken together support the conclusion that locally applied tacrolimus could be beneficial in models of lymphedema, the studies are somewhat superficial. Instead of carrying out the same experiments on several models, a more in depth analysis of mechanism of one or two models would have been more informative.

We thank the reviewer for reading our paper but respectfully disagree with this assessment. The various models are necessary to study different aspects of the pathology of lymphedema. For example, the hindlimb model is useful for studying physiologic changes that occur after lymph node dissection (similar to the clinical scenario) and enable us to study putative mechanisms by which tacrolimus is beneficial after lymphatic injury. For example, this model enables us to analyze collecting lymphatic pumping capacity. This is important since previous studies have shown that patients with lymphedema or who have undergone lymphatic injury have decreased lymphatic pumping.^{1,2} Our findings clearly

show that infiltrating T cells either directly or indirectly impair lymphatic pumping and that inhibition of this response is beneficial. These findings therefore not only provide a mechanistic explanation for why these changes occur clinically, but also provide a potential treatment option that has been used safely in other conditions. The hindlimb model is also useful for studying the mechanisms that regulate formation of collateral lymphatic vessels that bypass the zone of injury and our study provides strong evidence that this process is inhibited by infiltrating T cells. Although we can study this response with the tail model, the hindlimb is much more effective for this purpose since wound healing issues relating to excision of a full thickness portion of tail skin do not apply to the hindlimb. We support our findings in lymphangiogenesis in the hindlimb with well-accepted models of inflammatory lymphangiogenesis providing additional supportive mechanistic studies for our conclusions. The tail model, as used in our study, is useful for studying fibrosis and adipose deposition (these changes, similar to the clinical scenario, do not occur to an appreciable extent in the hindlimb model). Studying the role of T cells in regulating tissue fibrosis, collecting vessel fibrosis, and fibroadipose tissue deposition as performed in this study is important since these are the important clinical features of lymphedema. Our study therefore, provides strong evidence that T cells play a key regulatory role in this response and that topical inhibition of T cell responses can be therapeutic.

2. The authors' premise is that TH2 cytokines inhibit lymphangiogenesis and they should refer to the paper by Savetsky et al (2015) that stresses the anti-lymphangiogenic roles of IL4 and IL13. However, they do not measure TH2 cytokines after tacrolimus; thus the mechanism is still unclear. Why are IL4 and IL13 not analyzed in light of the authors' previous data?

We thank the reviewer for citing the paper published by our lab.³ We have analyzed IL-4 and IL-13 using ELISA and the results of these studies support our initial conclusion. These studies have been added to the results section (Figure 5J&K).

3. The authors study several different models of inflammation. In general, inflammation results in lymphangiogenesis, and even more lymphangiogenesis appears to be beneficial at least in the tail wounding model. How about in the cornea model (Figure 6?). Is the florid lymphangiogenesis induced by tacrolimus beneficial or harmful?

Lymphangiogenesis in the cornea is a pathologic condition and therefore not beneficial. However, this is somewhat irrelevant since we are not using the corneal model to study lymphedema, but rather as an assay to study the mechanisms of inflammatory lymphangiogenesis and how T cells contribute to this response. Our findings in the corneal model are supportive of our hypothesis that T cells are anti-lymphangiogenic since tacrolimus (a known T cell inhibitor) increased lymphangiogenesis.

4. What is the mechanism of lymphangiogenesis that is stimulated by tacrolimus? Is this proliferation of existing vessels, sprouting, generation of new vessels? Does this differ by model? The authors claim that tacrolimus results in lymphangiogenesis. However, in many cases, there do not seem to be more lymphatic vessels in the treated mice (eg. Fig 3A, 3C and 3D). Although the authors nicely show in figure 4 an enhancement of lymphatic vessel function (increased pulsing frequency, increased lymph node dye uptake) after treatment and increased collateral vessel density in the LN removal model after tacrolimus. More quantification of lymphangiogenesis (e.g. by proliferation by e.g. Ki67) would be appropriate.

Our findings in Figure 6A and 6F suggest that lymphangiogenesis during wound healing and corneal lymphangiogenesis occurs, at least in part, by sprouting of new vessels as reflected by increased branching. Our findings with the tail and hindlimb model also suggest that new lymphatics are generated providing collateral pathways for drainage (Figure 5A showing bridging lymphatic vessels). The representative images in Figure 3A-C and Supplemental Figure 4 are high-power images showing collagen deposition, TGF- β 1 expression, and pSMAD-3. Lower power images would show the differences in lymphatic vessel numbers however this would decrease our ability to show the pathological changes that we are discussing. The quantification of lymphatic vessel density is done at a lower magnification and is reflective of the number of lymphatic vessels present. We have also added Ki67 staining of lymphatics demonstrating increased LEC proliferation after tacrolimus treatment (Figure S10). It is hard to imagine that different mechanisms would be occur in different models for lymphangiogenesis since these mechanisms are, in general, well conserved. Therefore, based on our previous studies and the findings of other labs we hypothesize that tacrolimus stimulates inflammatory lymphangiogenesis by a variety of mechanisms including increased LEC proliferation, increased sprouting, and formation of new lymphatics by inhibiting T cell responses and decreasing the expression of anti-lymphangiogenic cytokines such as IL-4, IL-13, or IFN- γ . This hypothesis is supported by our findings demonstrating that tacrolimus treatment has no significant effect on the expression of prolymphangiogenic cytokines (e.g. VEGF-A or VEGF-C). However, this treatment markedly decreases the expression of anti-lymphangiogenic molecules (e.g. IL-4, IL-13, IFN- γ , TGF- β 1). Our hypothesis is also supported by the fact that tacrolimus treatment **ONLY** increased lymphangiogenesis in the setting of inflammation suggesting that this effect is an indirect function.

5. In supplemental Figure 10, the authors quantify LYVE-1+ staining area in controls and tacrolimus treated unwounded mouse ears. They need to carry out the same analysis in wounded mouse ears treated or not with tacrolimus.

This analysis was done and is shown in Figure 6D, E, and F.

6. Why does tacrolimus result in a decrease in lymphatic vessel pulsing intensity (Figure 4Di)?

Tacrolimus treatment results in increased lymphatic pumping frequency. The intensity shown in the graph is in arbitrary units and not quantified and will require additional analysis in future studies. Most likely this finding (if it in fact is a finding) reflects better clearance of dye from the hind limb thus decreasing the intensity of florescent staining.

Reviewer #2 (expert in lymphangiogenesis and lymphedema)

Remarks to the Author:

This is an elegant study illustrating the pro-lymphangiogenic effect of the T-cell inhibitor tacrolimus. The following issues should be considered.

Remarks

The authors should more consistently quantify other structural features of lymphangiogenesis besides lymphatic vessel density, such as lymphatic area, branching, morphology, etc in each model studied to support their conclusion that tacrolimus stimulates lymphangiogenesis.

We have analyzed lymphatic vessel areas in the hindlimb and consistent with our results demonstrating improved function have found that the lymphatic capillaries in tacrolimus treated mice have a smaller luminal diameter indicating decreased lymphatic vessel dilatation.⁴ We could not perform branch point analysis on histological sections of the hind limb since these were not whole mounts (in general, we have found it impossible to do this type of staining in hind limb skin). However, we did analyze branch points in the cornea and ear models and these have been added to our study (Figure 6C). The findings of these studies are supportive of our initial report.

1. An important control is missing: does tacrolimus affect (stimulate) lymphatic endothelial cells? A direct effect, independent of T cells, should be excluded.

This control was done (Supplemental Figure 7 and 11) and showed that tacrolimus treatment of normal tissues (i.e. non-inflamed normal skin) has no effect on collecting lymphatic pumping or in promoting lymphangiogenesis. As can be seen in supplemental figure 10, there is no difference in lymphatic vessel density, branching, or diameter in normal skin treated with or without tacrolimus. These findings, together with our findings showing that tacrolimus increases lymphangiogenesis in the setting of inflammation suggest that the effects of tacrolimus are indirect (i.e. decreasing anti-lymphangiogenic cytokines) rather than direct effects on LECs.

2. Tacrolimus has beneficial effects in the models used, but these are surgical models, in which inflammation is induced due to the surgery. Does tacrolimus also work in other lymphedema models?

Given that the vast majority of cases of lymphedema in developed countries are due to surgical injury as well as the fact that models of primary lymphedema are in most cases complicated by other pathological changes, we have chosen to study tacrolimus only in the setting of secondary lymphedema (with several different models). It is also important to note that a hallmark of lymphedema both clinically and in experimental models is progressive inflammation of the lymphedematous tissues even after the initial surgical wound has completely healed. In the models that we have used in this study, the wounds from the surgical excision are also completely healed when the tacrolimus therapy is initiated. Regardless, we have changed the title of our paper to “Topical Tacrolimus for the Treatment of *Secondary* Lymphedema” to avoid confusion. It remains possible that

tacrolimus may be effective in other lymphedema models but these studies will require dedicated manuscripts and additional work that is beyond the scope and space limitations of this journal.

3. The link between PLND and ROS is unclear and insufficiently characterized. The authors also did not show that "lymphatic injury results in generation of ROS" - this is an overstatement.

We have removed this analysis from the manuscript and plan to follow up this lead with a dedicated study.

4. Does tacrolimus affect blood vessels and their growth?

Tacrolimus had no effect on blood vessel growth and proliferation in the corneal model (Figure 6B). This is consistent with the finding that tacrolimus treatment did not significantly increase the expression of angiogenic growth factors (VEGF-A, VEGF-C).

5. The authors should support their conclusions by treating T cells in vitro with tacrolimus or control, and testing whether the conditioned medium affects lymphatic endothelial cells. Pro-lymphangiogenic cytokine activity in the conditioned medium should then be immuno-neutralized for the mentioned cytokines.

The mechanisms of action of tacrolimus have been well documented in a large number of in vivo and in vitro studies. These studies have shown that tacrolimus is a T cell inhibitor and our in vivo findings clearly support this role after lymphatic injury as well as in wound healing and the corneal lymphangiogenesis assay. We have previously published in vitro studies in which we and others have studied the effects of recombinant IL-4, IL-13, IFN- γ as well as neutralizing antibodies for these cytokines to study lymphangiogenesis.^{3,5} Thus, while we can activate T cells in vitro, then treat them with tacrolimus to inhibit them, and then neutralize the conditioned media with neutralizing antibodies against IL-4, IL-13, or IFN- γ in the current study, we feel that these studies would be largely repeating work that has been previously done and that we have published in our previous manuscripts. In addition, these studies do not take into consideration complex in vivo interactions that may occur with pro-lymphangiogenic cytokines produced by other cell types.

REFERENCES

- 1 Stanton, A. W., Modi, S., Mellor, R. H., Levick, J. R. & Mortimer, P. S. Recent advances in breast cancer-related lymphedema of the arm: lymphatic pump failure and predisposing factors. *Lymphat Res Biol* **7**, 29-45, doi:10.1089/lrb.2008.1026 (2009).
- 2 Modi, S. *et al.* Human lymphatic pumping measured in healthy and lymphoedematous arms by lymphatic congestion lymphoscintigraphy. *J Physiol* **583**, 271-285 (2007).
- 3 Savetsky, I. L. *et al.* Th2 cytokines inhibit lymphangiogenesis. *PLoS One* **10**, e0126908, doi:10.1371/journal.pone.0126908 (2015).
- 4 Tabibiazar, R. *et al.* Inflammatory manifestations of experimental lymphatic insufficiency. *PLoS Med* **3**, e254, doi:10.1371/journal.pmed.0030254 (2006).
- 5 Shin, K. *et al.* TH2 cells and their cytokines regulate formation and function of lymphatic vessels. *Nat Commun* **6** (2015).

Reviewers' comments:

Reviewer #1 (Remarks to the Author):

The manuscript by Gardenier et al evaluates the ability of tacrolimus, an inhibitor of calcineurin in T cells, to enhance lymphangiogenesis in various inflammatory, secondary, lymphedema models. In spite of the fact that lymphangiogenesis is associated with inflammation, previous studies had shown that TH1 cytokines (IFN γ) and TH2 cytokines (IL4, IL13) inhibit lymphangiogenesis. The authors' previous work concentrated on TH2 cytokines. The manuscript does not address that paradox concerning the increased lymphangiogenesis in inflammation, but does nicely show that T cell depletion can be associated with increases in functional lymphatics and decreased fibrosis after local treatment with tacrolimus. New data regarding decreases in IL4, IL13, and IFN γ support the conclusion that the effects are local and associated with inhibitory cytokine depletion. Appropriate controls demonstrating the absence of an effect of tacrolimus treatment on intact untreated ears, support the concept that this treatment could be useful in conditions of inflammation. The authors have provided new data purporting to show an increase in Ki67+ (dividing) cells in the vicinity of the LYVE-1 vessels. This suggests that the lymphangiogenesis that occurs after tacrolimus treatment is due in part to proliferating lymphatic endothelial cells, again most likely an indirect effect.

Reviewer #2 (Remarks to the Author):

General: Gardenier et al. report that topical administration of tacrolimus, an immunosuppressant that inhibits IL2 production and thereby reduces the T cell response, can be used for the treatment of lymphedema. They propose that the beneficial effects of tacrolimus are mediated by a T cell-dependent effect on lymphatic vessels, leading to enhanced lymphangiogenesis and improved lymphatic function. This study has benefitted from the revision, although a number of remarks still remain to be addressed.

Remarks:

General:

As requested, the authors supplied data on structural features of lymphatic vessels. However, a statement/data on the morphology of the vessels is still lacking. Did the authors observe any striking changes in tortuosity or thickness of the vessels? Any obvious irregularities in shape? In some of the representative images, it appears the increase in lymphatic vessels in tacrolimus-treated animals are primarily thin and sprout-like.

Comment 1:

The additional experiments performed by the authors involving tacrolimus treatment of non-inflamed models provides further information about its effect on lymphangiogenesis without the background proinflammatory signaling. Another important, more direct control for the implied mechanism (as already requested in the first round of revision) would be an in vitro experiment, treating primary lymphatic endothelial cells with tacrolimus and measuring the direct effects on proliferation and migration. This would exclude the possibility that there are direct effects on lymphatic endothelial cells, as in the mode of action of tacrolimus, the binding to FKBP12, a BMP repressor, relieves its inhibition of the BMP pathway, thus enabling BMP signaling via Alk1-3 (Dyer LA et al. Trends Endocrinol Metab, 2014). As BMP signaling has been reported to modulate lymphatic development/function (Dunworth WP et al. Circ Res, 2014; Kim J & Kim J. Mol Cells, 2014), this is of interest to better differentiate direct effects of tacrolimus versus T cell-dependent effects.

Comment 4:

It would be of interest to also quantify the effects of topical administration of tacrolimus on angiogenesis in the tail lymphedema model to differentiate from the effects of systemic administration of tacrolimus on angiogenesis shown in the corneal suture model.

Comment 5:

To answer this comment, the authors referred to literature and previous work. However, while the mentioned cytokines do inhibit lymphangiogenesis and the abrogation of their expression could account for the observed effects, it would be beneficial to contemplate about potential pro-lymphangiogenic cytokines, whose upregulation in tacrolimus-treated T cells could contribute as well. One such candidate could be, for example, IL8, which promotes lymphatic regeneration (Choi et al., *Angiogenesis*, 2013), and can be produced by T cells (Gibbons et al., *Nat Med*, 2014; Gesser et al., *Biochem Biophys Res Commun*, 1995). This could be done for example by ELISA-based quantification of cytokines in T cell-conditioned media upon tacrolimus treatment and subsequent characterization. If indeed a putative pro-lymphangiogenic cytokine may be altered, the original suggestion of a neutralizing antibody targeting this would provide proof-of-principle, mechanistic evidence of whether the enhancement in lymphangiogenesis is primarily via the inhibition of anti-lymphangiogenic cytokines versus the enhancement of pro-lymphangiogenic factors.

Minor comments:

1. Figure 2. The figure legend should be adjusted to (A-C) instead of (A-D).
2. Figure 4F. The number of LYVE-1 positive vessels in the image representing the control condition seems higher than in the tacrolimus treated condition. A more suitable and representative image should be chosen. Additionally, in the figure legend, the description of figure 4G and H (labelled as F and G) should be corrected.
3. Figure 4H: The labelling of the y-axis "INOS+ cells/hpf" should be clarified. What does "hpf" stand for?
4. Figure 5: The y-axis of graphs depicting IFN γ quantifications should be corrected from "INF γ " to "IFN γ ".

Response to reviewer's comments

Reviewer#1

The manuscript by Gardenier et al evaluates the ability of tacrolimus, an inhibitor of calcineurin in T cells, to enhance lymphangiogenesis in various inflammatory, secondary, lymphedema models. In spite of the fact that lymphangiogenesis is associated with inflammation, previous studies had shown that TH1 cytokines (IFN γ) and TH2 cytokines (IL4, IL13) inhibit lymphangiogenesis. The authors' previous work concentrated on TH2 cytokines. The manuscript does not address that paradox concerning the increased lymphangiogenesis in inflammation, but does nicely show that T cell depletion can be associated with increases in functional lymphatics and decreased fibrosis after local treatment with tacrolimus. New data regarding decreases in IL4, IL13, and IFN γ support the conclusion that the effects are local and associated with inhibitory cytokine depletion. Appropriate controls demonstrating the absence of an effect of tacrolimus treatment on intact untreated ears, support the concept that this treatment could be useful in conditions of inflammation. The authors have provided new data purporting to show an increase in Ki67+ (dividing) cells in the vicinity of the LYVE-1 vessels. This suggests that the lymphangiogenesis that occurs after tacrolimus treatment is due in part to proliferating lymphatic endothelial cells, again most likely an indirect effect.

Thank You

Reviewer #2

As requested, the authors supplied data on structural features of lymphatic vessels. However, a statement/data on the morphology of the vessels is still lacking. Did the authors observe any striking changes in tortuosity or thickness of the vessels? Any obvious irregularities in shape? In some of the representative images, it appears the increase in lymphatic vessels in tacrolimus-treated animals are primarily thin and sprout-like.

A statement on the morphology of the lymphatic vessels of tacrolimus treated animals was included in the manuscript Page# 11, line 12-13 and page# 12, line 7-9. No significant differences were observed in the lymphatic vessel thickness between control and tacrolimus treated ear punch wound margins (Supplemental Fig.11A)

The additional experiments performed by the authors involving tacrolimus treatment of non-inflamed models provides further information about its effect on

lymphangiogenesis without the background proinflammatory signaling. Another important, more direct control for the implied mechanism (as already requested in the first round of revision) would be an in vitro experiment, treating primary lymphatic endothelial cells with tacrolimus and measuring the direct effects on proliferation and migration. This would exclude the possibility that there are direct effects on lymphatic endothelial cells, as in the mode of action of tacrolimus, the binding to FKBP12, a BMP repressor, relieves its inhibition of the BMP pathway, thus enabling BMP signaling via Alk1-3 (Dyer LA et al. Trends Endocrinol Metab, 2014). As BMP signaling has been reported to modulate lymphatic development/ function (Dunworth WP et al. Circ Res, 2014; Kim J & Kim J. Mol Cells, 2014), this is of interest to better differentiate direct effects of tacrolimus versus T cell-dependent effects.

We have performed a series of detailed experiments to explain the direct effect of tacrolimus on LECs migration, tube formation and proliferation and found no significant difference with tacrolimus treatment as compared with controls. The data was included in Supplemental Fig. 12A-C and description was included in page# 12, lines 13-16 in the manuscript.

It would be of interest to also quantify the effects of topical administration of tacrolimus on angiogenesis in the tail lymphedema model to differentiate from the effects of systemic administration of tacrolimus on angiogenesis shown in the corneal suture model.

Blood vessel density from control and tacrolimus treated tail tissue sections were quantified as suggested by the reviewer. We found no significant differences between groups. Data is included in Supplemental. Fig.8C

To answer this comment, the authors referred to literature and previous work. However, while the mentioned cytokines do inhibit lymphangiogenesis and the abrogation of their expression could account for the observed effects, it would be beneficial to contemplate about potential pro-lymphangiogenic cytokines, whose upregulation in tacrolimus-treated T cells could contribute as well. One such candidate could be, for example, IL8, which promotes lymphatic regeneration (Choi et al., Angiogenesis, 2013), and can be produced by T cells (Gibbons et al., Nat Med, 2014; Gesser et al., Biochem Biophys Res Commun, 1995). This could be done for example by ELISA-based quantification of cytokines in T cell-conditioned media upon tacrolimus treatment and subsequent characterization. If indeed a putative pro-lymphangiogenic cytokine may be altered, the original suggestion of a neutralizing antibody targeting this would provide proof-of-principle, mechanistic evidence of whether the enhancement in

lymphangiogenesis is primarily via the inhibition of anti-lymphangiogenic cytokines versus the enhancement of pro-lymphangiogenic factors.

We thank the reviewer for this comment; however, there is only limited space and time for any particular question. This new question (new for the second review), we believe is beyond the scope of this study and an interesting question for future studies. Furthermore, contrary to the contention by the reviewer, previous studies have shown that phagocytes and mesenchymal cells are the major source of IL-8 rather than T cells (Baggiolini & Clark, FEBS Lett. 1992; Bickel, J Peridontol, 1993). Finally, although a few studies have shown a pro-lymphangiogenic role for IL8 in some settings, the source of this cytokine in general and the role of T cells in its production is highly debatable. The reference cited by reviewer (Gibbons et al., Nat Med, 2014) clearly states that IL-8 producing T cells are rare in adults and they have a main role only during neonatal stages.

Figure 2. The figure legend should be adjusted to (A-C) instead of (A-D).

Adjusted accordingly

Figure 4F. The number of LYVE-1 positive vessels in the image representing the control condition seems higher than in the tacrolimus treated condition. A more suitable and representative image should be chosen. Additionally, in the figure legend, the description of figure 4G and H (labelled as F and G) should be corrected.

Corrected accordingly

Figure 4H: The labelling of the y-axis “iNOS+ cells/hpf” should be clarified. What does “hpf” stand for?

“Hpf” stands for High power field. Essentially iNOS+ cells were counted from given high magnification images and plotted as a bar graph. However, for uniformity we have changed the Y axis labeling to iNOS+ cells (n) similar to Fig.4G

Figure 5: The y-axis of graphs depicting IFN γ quantifications should be corrected from “INF γ ” to “IFN γ ”.

Corrected